# The Effects of Social Presence and Familiarity on Children–Robot Interactions

**DOI:** 10.3390/s23094231

**Published:** 2023-04-24

**Authors:** Yi-Chen Chen, Su-Ling Yeh, Weijane Lin, Hsiu-Ping Yueh, Li-Chen Fu

**Affiliations:** 1Department of Psychology, National Taiwan University, Taipei 106216, Taiwan; 2MOST Joint Research Center for AI Technology and All Vista Healthcare, Taipei 106216, Taiwan; 3Center for Artificial Intelligence and Advanced Robotics, National Taiwan University, Taipei 106216, Taiwan; 4Graduate Institute of Brain and Mind Sciences, National Taiwan University, Taipei 106216, Taiwan; 5Department of Library and Information Science, National Taiwan University, Taipei 106216, Taiwan; 6Division of e-Learning, Computer & Information Networking Center, National Taiwan University, Taipei 106216, Taiwan; 7Department of Bio-Industry Communication and Development, National Taiwan University, Taipei 106216, Taiwan; 8Department of Electrical Engineering, National Taiwan University, Taipei 106216, Taiwan; 9Department of Computer Science and Information Engineering, National Taiwan University, Taipei 106216, Taiwan

**Keywords:** social presence, familiarity, rapport, repeated children–robot interaction, negative attitudes toward robots

## Abstract

In children–robot interactions, an impression of a robot’s “social presence” (i.e., an interactive agent that feels like a person) links positively to an improved relationship with the robot. However, building relationships takes many exposures, and there is an intellectual gap in how social presence and familiarity collaborate in modulating children–robot relationships. We investigated whether social presence altered over time, how repeated exposure and social presence affected rapport, and how social presence would modulate children’s attitudes toward the robot. Fourteen children (four female, age = 10.79 ± 1.12) interacted with a companion robot for four days in spontaneous interactions. The findings revealed that children who perceived the robot as having a higher social presence developed a stronger rapport than those who perceived a lower social presence. However, repeated encounters did not change the children’s perceptions of the robot’s social presence. Children rated higher rapport after repeated interactions regardless of social presence levels. This suggests that while a higher social presence initially elevated the positive relationship between children and the robot, it was the repeated interactions that continued solidifying the rapport. Additionally, children who perceived a higher social presence from the robot felt less relational uneasiness about their relationship with robots. These findings highlight the importance of robots’ social presence and familiarity in promoting positive relationships in children–robot interaction.

## 1. Introduction

As technology advances, robots’ characteristics and skills will improve significantly, increasing human society’s reliance on their services. Many experts from various technology fields believe that robotics will soon be a common sight for the general public [1].

Children are probably the first generation of users to grow up in the era of adopting robots in everyday household tasks [2]. It is unclear, however, what encourages them to accept robots as social partners. Furthermore, there is a dearth of knowledge of what would deter children from interacting with robots and much less evidence regarding what could alter such negative concerns about robots.

In this study, we proposed two critical social factors that facilitate interpersonal relationships as candidates to explain the development of children–robot relationships. The first factor is social presence, a psychological representation of how much a robot is considered a social agent. We argued that individual differences in social presence are critical in explaining rapport development, characterized as a relationship with mutual understanding and affinity [3,4,5]. In addition, we also examined how children’s familiarity with the robot came into play in developing rapport with the robot. We then examined how a robot’s social presence modulates children’s concerns toward robots after familiarizing themselves with the robot. In this regard, we aimed to shed light on specific aspects of children’s negative attitudes about robots and explored if interacting with a sociable robot could help lessen their worries.

### 1.1. The Perceived Social Presence of Social Robots

Social presence is the degree to which users perceive an interactive agent as a “real person” [6]. During communication, social presence emerges when people feel they have access to another entity’s thoughts and emotions [7], not mediated by others [8,9,10]. The concept originated from mediated communications between humans (e.g., online meetings), where the interlocutor could be presented more or less in the interaction. Therefore, social presence is a continuous psychological perception of a given social partner than a binary concept of “being” or “not being here” [6,11]. Just as a recent study concluded, “the more social cues robots display, the more competent they are judged to be by adults” [12]. A stronger perceived social presence thus induces humans’ greater reliance on interpersonal social scripts during interactions with a virtual agent [13] or robot [14].

Social robots, designed to perform human-like interactions, are intended to display a social presence through their humanoid appearance and autonomous behaviors [15,16,17,18,19]. Critically, the existing literature suggests that children’s social cognition displays significant variability based on the developmental stages. For example, 5-year-old children would attribute a humanoid robot alive at chance level [20]. On the other hand, children aged 7–9 have increased their sensitivity to robots’ social presence given the different anthropomorphic features (e.g., mechanical and humanoid [16]). Moreover, the likelihood of treating the robot as a social entity is significantly reduced as children become adolescents [21].

Social cue sensitivity in human–computer interactions (HCIs) explains why people’s perceptions of social presence vary with the same computer agent [11]. As a result, given that children’s social awareness varies greatly during different developmental stages [22], their contact with robots may be met with various baseline attitudes.

### 1.2. Perceived Social Presence Facilitates Children–Robot Relationship

Social interaction, the process involving reciprocal stimulation or response between individuals, is emotionally rewarding to humans by nature [23]. Pleasure during social interactions is the main incentive that drives children to interact with a social robot [24]. Hence, social robots are widely used as children’s tutors [25,26,27], coaches [28,29], and even companions providing emotional comfort [30,31]. In children–robot interactions (CRIs), social robots usually provide their services to children through gamified activities and are introduced as playmates, providing guidance and support during tasks. Compared to computer-based virtual agents, children reported being more engaged when interacting with social robots with physical embodiments [27,32].

Consistently, increasing the sense of agency has been positively linked to robots’ impressions. For example, robots with a higher perceived social presence seemed more attractive and intelligent to children [33]. Furthermore, substantial research shows that the interactive outcomes after one-time children–robot interactions increased as robots achieved a greater social presence, such as enjoyment of interaction [34], improved learning ability [35], increased positive mood [30], and a higher acceptance of robots [2]. Children also exhibited an increased exchange of social cues, a behavioral indicator of rapport [36], during interactions with robots that were perceived to have a more potent social presence.

Specifically, Heerink et al. showed that a robot with nonverbal communicative abilities (e.g., making animal sounds and body movements) was sufficient for children to assign social presence to it. In the same study, children would initiate behaviors critical for rapport building, such as teaching new skills and requesting attention from the robot [33]. Another study also suggests that physical play is likely the source for children to perceive social presence from the interaction [21].

Overall, current evidence suggests that a robot’s social presence, manifested through its hedonic features, could greatly contribute to children’s decision to treat it as a social partner. In light of this, individual variations in perceiving social presence tend to be the prominent factor in explaining children–robot relationships.

### 1.3. The Interaction Effect between Social Presence and Familiarity on Children–Robot Relationships

Generally, the longer people interact, the more familiar they are with their partners, resulting in a stronger rapport [37]. Rapport, the feeling of being “in sync” with one’s partner in a conversation, is argued to underlie many desirable social outcomes, such as having a high emotional quality interaction [38] and increased self-disclosure, cooperation, liking, and affiliation [3,4,5].

However, there is also a situation where repeated encounters do not necessarily guarantee a closer relationship. For example, while testing a social robot in the domestic setting, de Graaf et al. found that adults who sustained using the robot for over six months reported a firm psychological attachment to it. By contrast, another group of users with greater initial expectations of the robot showed little attachment to it. They stopped using the robot as soon as they learned it failed to achieve their initial expectations of its functionality [39]. Suggestively, familiarity’s effect on human–robot relationships differs among users who held high versus low impressions of the robot partner.

As mentioned earlier, the children’s social cognition of robots changes drastically within the age range of 5–12 (e.g., [16,20,21]). However, except for one study [40], such a phenomenon has rarely entered the scope of existing CRI research investigating the interaction outcomes. Hence, it is unknown if the variation in social presence affects familiarity in children–robot relationships. Against the backdrop, we aimed to explore whether children who perceived a high versus low social presence from a social robot would develop different levels of rapport with the robot over time. Moreover, we also examined if familiarity’s effect on rapport differs between children who perceived a high versus low social presence, as [39] has suggested in adult users.

### 1.4. The Perceived Social Presence on Children’s Negative Attitudes toward Robots

In addition to positive appraisals, users can sometimes experience negative emotions during HRIs. The coexistence of negative attitudes with positive evaluations of robots is not uncommon. For instance, although adult users prefer a mindful AI robot to a fallible one, they are also wary that it becomes out of humans’ control [41]. Negative feelings, such as anxiety and fear about social robots, could arise from beliefs before the interactions, such as worries about information privacy in general [42] and the mental image of robots’ humanness [19]. Although in-person interaction experiences promote positive feelings, this does not seem to offset negative attitudes toward robots [43]. Consistently, a recent study has shown that for adults, pre-interaction bias toward robots is hard to alter [44].

Research on robots’ utilities rarely directly assesses children’s negative feelings and anxieties about interacting with robots, most likely because of their relatively permissive attitudes toward robot malfunctions. Among the handful of studies, one found that younger children showed no changes in the robot’s likability and remained willing to interact with a robot even when it occasionally violated the expectation to keep information private [45]. On the other hand, [46] believes that robots that lack social awareness and show emotion inappropriately in the situation may cause children to feel uneasy and scared. Although these implications are insightful, a lack of direct exploration of children’s negative emotions has left gaps in explaining the mixed findings of children’s concerns about robots.

### 1.5. Research Questions

Hence, intriguing questions emerge in the direction where current evidence converges. In this study, we asked the following research questions in the current study:Does a robot’s perceived social presence enhance or decrease after children have become familiar with the robot?

Our experimental design allows us to see if the perceived social presence changes as children become more familiar with the robot. Therefore, we tested this question by comparing the social presence ratings of the robot between interaction sessions—shortly after the first vs. the last CRI.

2.How do children’s impressions of a robot’s social presence and progressive familiarity with the robot affect the rapport-building process?

To answer this question, we measured children’s self-reported rapport with a robot in the middle of and after several interactive sessions.

3.Can the levels of social presence children perceive from a robot modulate their pre-existing bias toward robots?

Because individuals could develop a bias toward robots before they have actual interaction experiences with robots, we collected children’s negative attitudes toward robots before starting and right after interactions. This allowed us to determine if interacting with a robot expressing a sense of social presence could modulate the negative attitudes children developed before this interaction.

## 2. Materials and Methods

### 2.1. Participants

Children aged 6–12 who attended a five-day robot programming workshop hosted by the Computer and Information Networking Center at National Taiwan University, Taiwan, were invited to participate in this study.

In the five-day robot programming workshop, these children learned to control Zenbo, a personal robot, to perform basic operations (such as moving around, taking photos, and broadcasting internet search results) through Scratch, a high-level block-based visual programming language.

After describing the experimental content to the children and their parents, 19 children (6 females; age = 10.47 ± 1.47) enrolled in the study and supplied signed consent forms from their parents. Five children did not complete the experiment due to their request to end the interaction, absence from the workshop, or inability to understand the task. The data from these children were thus excluded, leaving a final sample of 14 participants (4 females, age = 10.79 ± 1.12).

### 2.2. Socially Interactive Robot

A small humanoid robot, RoBoHon (Sharp Co., Ltd., Osaka, Japan), was introduced to interact with the children. RoBoHon, as a commercial robot, has reasonable speech capabilities and can generate body gestures by moving its head and arms. In addition, because it has a database of pre-programmed social interactions, it was selected for the current human–robot interaction study.

Given that social interactions are the exchange of responses between interlocutors, we defined actions that (1) demand children’s attention and responses and (2) react appropriately in a given context as social behaviors. Given that our focus was the relationships between these two purposed factors, we adopted social activities extensively utilized in CRI research due to their well-established effects on positive children–robot relationships. We used six activities in the iterating system: joke, riddle, chit-chat, tale, dance, and video. Children engaged in the interactions by guessing the punchlines, listening to riddles and tales, chit-chatting, and watching dances and videos initiated by RoBoHon.

Each activity category has averaged 10 different sets of actions. RoBoHon would not repeat the same single action twice to the same user. With basic Natural Language Processing (NLP) capabilities, RoBoHon could autonomously generate adaptive actions based on the children’s responses. For example, if asked, “would you like to hear a humorous joke?” a yes response would initiate a joke-telling session, whereas a no response would initiate another activity proposal.

The participants’ identities were predefined (e.g., name and gender) and used in the robot’s utterances in the subsequent interactions. To ensure the confidentiality of personal information, participants’ identifications were then replaced by number codes as data identifiers and saved in an encrypted cloud driver that was only accessible to the research team.

### 2.3. Measurements

We introduced a set of instruments that measure the attitudes toward social robots at the three-time points during the interaction sessions:Pre-interaction (T_0_): Before interacting with RoBoHon, the questionnaire collected reflected the children’s negative attitudes toward robots.Mid-interaction (T_1_): At the end of the first half of the experimental session, with approximately 15 min of accumulated interaction time, the children’s first-time social presence and perceived rapport with RoBoHon were evaluated.Post-interaction (T_2_): After finishing all the interactive sessions, with around 30 min of accumulated interaction time, the perceived social presence, rapport, and the children’s negative attitude toward robots were evaluated. The differences between first-time and post-interaction for each evaluation reflected the attitude change after interacting with RoBoHon multiple times.

The reliability of these instruments was established by calculating Cronbach’s α to measure the internal consistency; the instruments above the recommended level of 0.7 were considered reliable, and their results would be used for further statistical analyses.

#### 2.3.1. The Negative Attitude toward Robots

The Negative Attitude toward Robots (NARS) scale measures individuals’ negative attitudes toward robots [47,48], with 11 items reflecting three dissociable concerns about robots: future/social influence (FSI, three items, T_0_: α = 0.57, T_2_: α = 0.53), actual interactions and situations (AIS, three items, T_0_: α = 0.17, T_2_: α = 0.81), and relational attitudes (RA, five items, T_0_: α = 0.62, T_2_: α = 0.75). One item in the RA subscale identified as contributing to the diverse response was the third item: “I feel anxious when talking to the robots”. After deleting the item, Cronbach’s α was 0.71 and 0.79 for T_0_ and T_2_, respectively. However, FSI and AIS failed the reliability test since no modifiable items were found. As a result, we eliminated the two subscales from further analysis.

#### 2.3.2. Social Presence

Bailenson et al. first developed the 5-item social presence questionnaire to examine how mutual gaze behavior mediates the sense of agency to an interactive virtual agent [49]. The questionnaire was later adapted by Herrink et al. for CRI and used to assess the children’s sense of being in the presence of an actual social entity with the feeling that it was not mediated by others [33]. Based on the purpose of the current children’s study, we used the version of [33] with a score ranging from 5 to 25 (T_1_: α = 0.86; T_2_: α = 0.83).

#### 2.3.3. Rapport with RoBoHon

The Human-Agent Rapport Questionnaire (HARQ) was adopted to measure the perceived children–robot relationship after free interactions [36]. The HARQ is a one-factor scale with 14 items (T_1_: α = 0.90; T_2_: α = 0.83).

### 2.4. Procedure

Due to limited time slots, we divided the CRI into four interactive sessions for four consecutive days, each running in three groups simultaneously. Each CRI lasted 7 to 8 min daily during the break times of the children’s robotic camp (from day 1 to 4). Overall, the interactive time with RoBoHon totaled around 30 min for participants who completed the experiment. All participants were randomly assigned into one of three groups and introduced to the same experimenters they would meet in the following days. These three groups of experiments were held in different rooms with identical room setups (see Figure 1) and experimental protocols.

At the beginning of the study, an experimenter and a technician introduced themselves to each child. Known to the children, the technician who sat behind a monitor was there to help the RoBoHon run. Only when the robot made speech recognition errors (e.g., misunderstanding the participants’ speech) would the technician rewrite the verbal inputs to ensure the conversation proceeded smoothly. Otherwise, RoBoHon functioned fully autonomously in the CRIs. During interactive sessions, children could chat freely with RoBoHon, while the robot would perform actions according to children’s responses (see section Socially Interactive Robot for more details). All the children were fully aware that they could end the interactions at any time without giving a reason.

At the following three time points of the CRI, T_0_, T_1_, and T_2_, the experimenters interviewed the children using questionnaires (see Section 2.3).

### 2.5. Analysis

First, we investigated whether social presence changed over time by measuring the first-time social presence (SP, measured at T_1_) and repeated CRIs (measured at T_2_). To this end, we grouped the children into High and Low SP groups based on all participants’ overall mean of first-time social presence.

We then modeled the development of rapport under the given SP group during repeated CRIs using a two-way mixed analysis of variance (ANOVA) on HARQ (between-subject factor: Low vs. High SP groups; within-subject factor: T_1_ vs. T_2_). Subsequently, a two-way mixed ANOVA was applied to test whether the changes in social presence across repeated CRIs (if any) mirror the rapport development.

Finally, we explicitly focused on whether the extent of social presence, grounded in the interactions with RoBoHon, could modulate the children’s negative attitudes toward robots. In this respect, we applied a one-way analysis of covariance (ANCOVA) to test whether social presence modulates post-interaction negative attitudes by comparing the NARS at T_2_ between different SP groups while controlling children’s prior bias (i.e., NARS at T_0_) towards robots.

## 3. Results

### 3.1. Social Presence

We compared attitude evaluations among two groups of children rated high versus low first-time social presence when meeting the robot (Low SP: Mean = 15.3 ± 2.2, High SP: Mean = 22 ±1.9). An independent-sample t-test showed that age did not differ among the two groups (Low SP: M = 11.14, SD = 0.9; High SP: M = 10.43, SD = 1.27; t(12) = 1.21, *p* = 0.25).

The ANOVA analysis showed no significant interaction between the Low vs. High SP group and T_1_ vs. T_2_ [F(1, 12) = 1.33, *p* = 0.27, ηp^2^ = 0.10] (Figure 2). The main effect of T_1_ vs. T_2_ was not significant [F(1, 12) = 0.85, *p* = 0.37, ηp^2^ = 0.07], indicating that the social presence assessed between the two-time points (mid- and post-interaction) did not differ for both groups. We found that the main effect was between the SP groups [F(1, 12) = 44.51, *p* = 0.00, ηp^2^ = 0.79]. This suggests that children who evaluated the robot with a higher first-time social presence maintained this perception even after repeated CRIs. Accordingly, the children’s perceived social presence was independent of their increased familiarity with the robot.

### 3.2. Rapport

The results suggested no interaction between the Low vs. High SP group and T_1_ vs. T_2_ on rapport [F(1, 12) = 0.23, *p* = 0.64, ηp^2^ = 0.02] (Figure 3). Furthermore, the main effect of social presence on HARQ showed that the rapport was marginally higher in the High SP than in the Low SP group [F(1, 12) = 4.69, *p* = 0.051, ηp^2^ = 0.28]. Specifically, the main effect of T_1_ vs. T_2_ [F(1, 12) = 9.67, *p* =.009, ηp^2^ = 0.45] suggests that the children–robot rapport increased after a series of CRIs.

### 3.3. Negative Relational Attitudes (RA)

The results of the ANCOVA showed that the covariate, pre-interaction RA (T_0_), was marginally significantly related to children’s post-interaction RA (T_2_) [F(1, 11) = 4.37, *p* = 0.061, ηp^2^ = 0.28] (Figure 4). There was a significant effect of the SP group after controlling for the pre-interaction RA [F(1, 11) = 12.62 *p* = 0.005, ηp^2^ = 0.53]. After the interactions, the children who perceived a higher social presence during the CRI reported a lower negative relational attitude towards robots than those who perceived a lower social presence.

## 4. Discussion

Growing evidence shows that more positive interactive outcomes are achieved when robots are perceived with a stronger social presence in single-session interactions [2,28,30]. At the same time, after becoming acquainted with a robot, an initially more favorable impression may result in a decreased rapport for adult users [39]. However, such a phenomenon has rarely been studied in children–robot interactions.

Thus, we first investigated the stability of perceived social presence. Subsequently, we examined the impacts of a robot’s social presence and familiarity with the robot on children–robot rapport. We then explored how this interactive experience modulates the children’s negative attitudes toward robots. Specifically, we tested if the levels of a robot’s perceived social presence could alter children’s pre-existing bias toward robots.

### 4.1. First-Time Social Presence Remained Stable after Children Became Familiar with the Robot

Our results suggest that the social cues from RoBoHon were sufficient to sustain children’s perceptions—regardless of the extent—that the robot was their social partner. The social robot in our study, RoBoHon, was programmed to use the child’s name in every interaction. Additionally, RoBoHon would adapt its response to the child if they rejected the prior ones during interactions (e.g., “What should we talk about then?” if they did not want to hear a joke). These convey a personal and respectful interactive style that enhances children’s engagement with the robot [29]. Thus, this interactive style could be essential for triggering a considerable social presence. Overall, despite the levels varying among individuals, the perceived social presence from RoBoHon satisfied the children’s expectations and maintained their motivation for interactions.

Although children have been known to have a more stable social attachment to objects than adults [51], age is not the only factor in maintaining a social presence over time. For example, after five weeks of interaction, the children felt a reduced social presence from a chess robot [52]. Later, the same research team was able to keep the robot’s social presence from diminishing over the same period of chess play (i.e., five weeks) by simply adding an empathetic module that allowed the robot to cheer and provide emotional support while playing chess [31]. These results align with a previous finding that children were generally more attracted to and willing to interact with robots that demonstrated socially adaptive behaviors, such as changing vocal and nonverbal output in response to children’s answers [2].

### 4.2. Regarding Rapport Building, Children’s Impressions of a Robot’s Social Presence Did Not Interact with Familiarity

After four days of interaction, all children’s rapport with RoBoHon improved. The strengthened rapport is partially explained by the classical mere exposure effect, in which repeated exposure to a stimulus increases one’s positive perception toward it [53]. According to a previous study [54], repeated exposure to an object eliminates human uncertainty, resulting in more efficient perceptual processing speed—indicating increased familiarity with the object—which relates to positive affect. Similarly, children playing with a robot for up to two weeks reported significantly improved feelings of ease toward their robot partner [55]. In addition to the ease, children’s friendly gestures also increase over time. For instance, children tended to imitate their robotic partner’s machinery behaviors (e.g., adopt the very slow and “stiff” nodding as the robot did) as the interaction time increased [55], an iconic behavioral synchronization that is commonly found among close friends [56].

The robot’s first-time social presence had a marginally significant effect on boosting the overall rapport with the children. Additionally, this advantage was stable across the repeated CRIs, unlike the interaction effect observed in adults [39]. Based on de Graaf et al., there were two distinct patterns of rapport building between adult users and the social robot. The first type of utility-driven users stopped interacting with the robot when they found it did not meet their high expectations for functionality. The second type of users were those who used the robot regularly and reported having a profound emotional attachment to it. The latter, who accepted the social robot in long-term interactions, did not have high anticipation in terms of the robot’s ability at the beginning of the experiment. The results indicate that the level of adults’ first impression of a robot moderates human–robot relationships differently over time.

The inconsistent evidence between adults and children raised an alternative explanation for the interaction effect. Presumably, the essence, instead of the extent of expectations for robots, is more critical in leading the directions of human–robot relationships. In other words, only those who build expectations mainly upon the robot’s utility would experience the gap between expectation and reality. As such, they initiate performance-based evaluations primarily during HRI, of which the standards are strictly predefined (e.g., the robot should understand everything I said). A violation of the users’ expectations thus leads straight to a verdict of incompetence on the robot. Users would no longer focus on the robot just as an interviewer rejects an unqualified candidate.

On the contrary, individuals with a rather vague image of robots and willing to interact with them tend to be surprised whenever they learn a robot’s new ability; this is analogous to the beginning of a friendship when friends are not actively expecting to benefit from the relationship. Unless interpersonal conflicts, boredom, or disgust develop, the more one becomes familiar with their partner, the stronger their connection will be [57]. This latter pattern aligns more with children’s relationships with the social robot. Even with preconceptions about the robots’ capability, children were not noticeably troubled when robots failed expectations [58,59]. In this case, a higher initial image of social presence does not necessarily lead to greater disappointment in children. Instead, the robot’s stronger social presence has provided an elevated starting point to further building rapport with children.

Our results suggest that social robots trigger a stable sense of social presence in children, which plays a crucial part in their decision to form an instant rapport with robots. Hence, children’s variations in attributing social presence to a robot explain their differences in their rapport development. Nonetheless, by sufficient interaction, it is possible to elevate children’s social bonding with a robot, which started low, given its not initially impressive social presence. On a broader scale, our research sheds light on the mechanism of the children-rapport building by showing that the extent to which social presence and familiarity contribute to children–robot rapport varies across temporal dimensions.

### 4.3. The Increased Social Presence Attenuated Children’s Pre-Existing Negative Attitudes toward Robots

After controlling for their before-interaction biases of robots, the children who perceived a higher social presence rated a lower relational (negative) attitude than those who perceived a lower social presence. Similarly, a recent review by Naneva et al. showed that for adult users, interacting with a robot face-to-face elicited more positive feelings or emotions toward robots than some form of indirect interaction (e.g., watching a video). However, compared to indirection interactions, face-to-face interactions also contributed to more significant concerns about the robots’ functions as the chances of exposing the robot’s flaws have increased [43]. Therefore, the authors conclude that the affective and cognitive concerns regarding robots are partially dissociable.

Our findings, extended from this account, help shed light on the mechanism of children’s attitude changes after direct interactions. Children’s interactive experiences with a robot modulated their relational uneasiness (i.e., RA score) toward robots. Significantly, the stronger the hedonic-related social presence they perceived, the greater the improvement in uneasiness. This result is also compatible with the view that social presence is a product shaped by the affective components that one senses during interactions [51], which could be more relevant in shaping children’s negative attitudes about robots than the cognitive concerns of utility (e.g., [58,59]).

Hence, our findings highlight that the underlying mechanisms of children’s negative attitudes toward robots could differ from those of adults. The degree of a robot’s social presence is critical for children in shaping their negative attitudes toward robots.

## 5. Contribution

Humans are hardwired to unconsciously detect and process social cues [60], evidenced by the easiness of anthropomorphizing animals and still-life objects [51]. Children at the developmental stage of practicing social skills are specifically prone to assigning mental activity to non-human beings [51]. Growing up in this digital era, children in this generation are inevitably more exposed to robots than their parents were in their childhoods. Therefore, it would be natural for children to consider a companion robot as a toy and interact with it with the social rules they learned through observing the caretakers’ interactions.

Previous work indicates that children’s impression of a robot’s social presence varies significantly among each other [16,20,21] and is positively related to the instant children–robot relationships [2,28,30,33,34,61]. However, how variation in social presence affects relationships have been largely overlooked by long-term CRI research (e.g., [17,55,62]). Extended from this, we found that in addition to increased familiarity, a robot’s perceived social presence is critical in producing a long-term rapport.

Additionally, we showed that children are not disappointed in robots even after familiarization. Evidence suggests that adult users who initially hold a strong impression of the robot would experience an attitude plummet after repeated interactions [39]. On the contrary, children’s first impression of a robot’s social presence is relatively stable and predictive of their relationship.

Moreover, our results indicate that the weighting of rapport-related factors should be adjusted according to application scenarios of robotic services. For instance, robots with a more substantial social presence are better suited for emotional comfort or hospitality services where instant rapport is needed. On the other hand, in long-term learning environments, tutor robots with a moderate social presence may be preferred, as robots with overwhelming social abilities may distract children from their current learning content [18]. In this case, the rapport that keeps children motivated in learning would be better accumulated through repeated lessons rather than an overwhelming social presence.

It is crucial to note that our research did not only consider the bright side of children–robot relationships. As far as we know, this is the first CRI research to systematically measure negative attitudes other than utility-based attitudes, such as trust [58,63,64]. For example, children’s negative attitudes toward robots indicate they do not fully view robots as harmless toys. In a similar vein, recent neuroscientific evidence indicates that humans perceive robots as distinct from mere objects and human beings [65]. Hence, it is worth mentioning that rapport in CRIs is not necessarily equivalent to that originating from human–human interactions. With the awareness of these differences, we hope our study could sparkle further investigation for more careful consideration of the underlying mechanisms of children–robot relationships.

From a larger scale, our findings provide a novel perspective on CRI research. Our findings suggest that researchers should consider variations in individuals’ attitudes (i.e., perceived social presence) and temporal metrics (i.e., repeated interactions) for designing robots’ affective models for children. This approach can contribute to developing social robots that simulate rewarding perceptions through interactions that foster long-term human–robot relationships. Additionally, apart from the advantages of robots, exploration of negative affect can aid in avoiding the adverse outcome of children–robot interactions, which may be the most significant contribution of such research.

## 6. Limitations and Implications

Some caveats should be mentioned about our current attempt to understand the effects of social presence in children–robot relationships. First, as social presence is a subjective perception based on the specific interaction context [51], one should be cautious when comparing HRI research that uses different robots in different circumstances. For example, in a home use context, some adults reduced their rating of a robot’s social presence over time [39]. Nonetheless, the social presence remained stable for all adults when playing an economic game with a robot (e.g., repeated prisoner dilemma) [66]. The apparent paradoxical results could be owing to variations in social contexts. For example, the former robot was introduced in unscripted interactions and presented with various functions. On the other hand, the latter was portrayed as another player capable of evaluating chess plays and making decisions. Different social contexts might thus direct the participants’ attention distinctively when evaluating performance (e.g., on a robot’s capability or reaction to one’s move in the game).

The spontaneous interaction setting in our study is analogical to the contexts introduced by de Graaf et al. [39]. Still, we stressed the companionship characteristics in the interaction more, which could inevitably attract children’s focus to the robot’s playful features. Collectively speaking, playing is a general approach taken by most CRI research to keep children engaged in the interactions, and even tutoring robots are usually introduced as playmates [26,27,28,29,30,31,32,33,52,62]. Hence, we recognized that maintaining social presence under a companion-focused context could be easier since the participants’ attentions are rather confined to certain features.

This study did not address how individual differences in digital acceptance and proficiency may affect children–robot relationships. The children in our study were recruited through a robot camp held during spring break, and their participation in the camp may indicate a higher interest in robots than their peers. Additionally, being able to participate in the robot camp could reflect differences in caretakers’ expectations of technology acquisition. For example, parents’ ambivalent attitudes about the robot’s ability could hinder the children’s adoption of robots [67]. As a result, children’s interests and families’ liberal attitudes toward robots could promote more positive attitudes toward a given robot.

Experiences with technology, or digital literacy, have been suggested to explain people’s expectations of robots. Compared to those with more digital proficiency, less digitally literate people may express lower expectations regarding robots’ functionality and perceive social robots as more novel items or toys [44]. Similarly, children from a developing country with limited exposure to robot services were more impressed by a social robot. In addition, they enjoyed playing with the robot more than the children from a developed country [34], suggesting that relationships with social robots may closely tie to the individual variations of digital proficiency and cultural backgrounds.

Moreover, there was insufficient internal reliability from other NARS subscales, resulting in the absence of analysis of children’s cognitive attitudes about robots (i.e., the FSA and AIS subscales). This could manifest the probability that children’s attention had focused on the robot’s affective features over cognitive ones during interactions, as suggested by CRI studies [24,68]. Alternatively, it could reflect children’s limited understanding of some items in NARS. For instance, the item “I would hate the idea that robots or artificial intelligence were making judgments about things” could be a description that is too abstract for children to conceive.

### Future Directions

In summary, to advance our current knowledge of children–robot relationships, we suggest the following directions for future research:

First, perceived social presence is sensitive to the characteristics of social partners and their interaction settings. As a result, comparing the perceived social presence of different robots in a given scenario, or exploring the changes in social presence across different social contexts with the same robot, would help shed light on modeling social presence and its impacts on relationships.

Second, future research should include more temporal parameters for CRIs, such as manipulating frequent versus occasional interaction frequency with short versus long interaction length. These issues aid in advancing our understanding of optimal children–robot relationships (which could vary by interaction goals).

Third, digital acceptance and literacy, which children could inherit from families and cultures, are involved in establishing expectations about social robots. Hence, investigating how children’s expectations differ based on their attitudes and digital background toward social robots could be a promising avenue for future research. Future studies should also consider that cultural diversity, such as variations in races and languages, could influence attitudes and behaviors toward social robots.

Finally, it is imperative to develop linguistically and cognitively sound instruments to gauge children’s concerns about robots. Nowadays, children’s worries about robots are sparse in the CRI literature, which may be owing to a paucity of appropriately prepared instruments on the subject. Nevertheless, our study demonstrated that children expressed negative attitudes toward robots, especially in relational aspects. Hence, further research is needed to identify children’s negative attitudes toward robots and the effects of these attitudes on the child–robot relationship.

## 7. Conclusions

The extent to which people perceive social presence from a robot is subject to their social sensitivity, which varies among individuals. Our study has shown that such variations can give some children a head start in building rapport with a robot. However, in terms of cementing the relationships, what is essential is the sense of familiarity acquired through repeated interactions. On the other hand, the benefits of the robot’s higher social presence are not confined to the current relationship but contribute to reducing children’s general relational uneasiness toward robots.

## Figures and Tables

**Figure 1 sensors-23-04231-f001:**
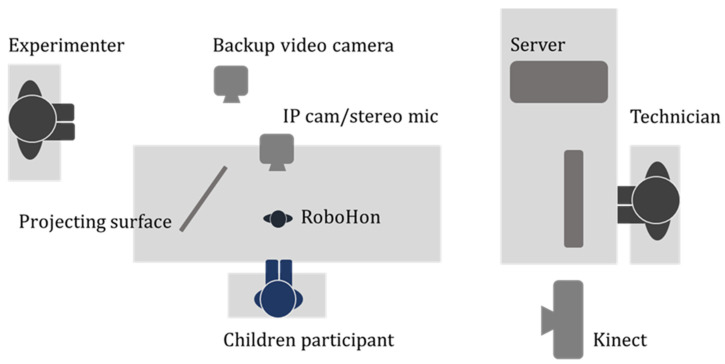
The experiment setup. The experimenter was the one who conveyed instructions for evaluations and answered all the inquiries from participants. The technician who maintained communication between the internet server and RoBoHon did not communicate verbally with the participants. The projecting surface was where the RoBoHon displayed videos. For data recorded by the IP camera, backup camera, and Kinect, please find [50] for further details.

**Figure 2 sensors-23-04231-f002:**
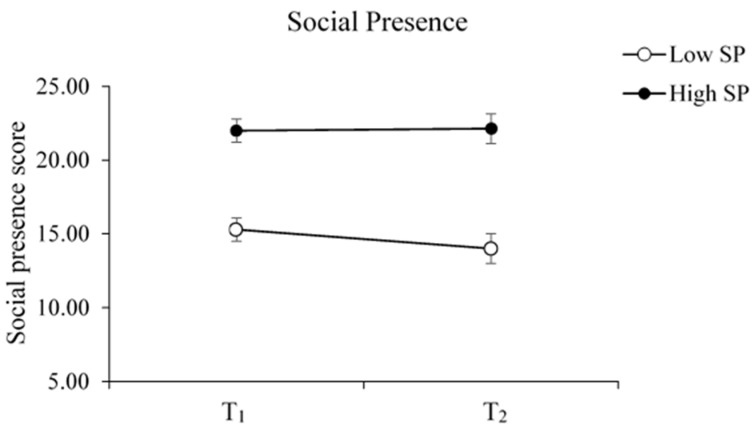
The change in social presence across different groups after repeated CRIs. The children decided on a robot’s social presence at their first meeting (T_1_), and this perception was stable across repeated CRIs. Error bars represent one standard error from the mean.

**Figure 3 sensors-23-04231-f003:**
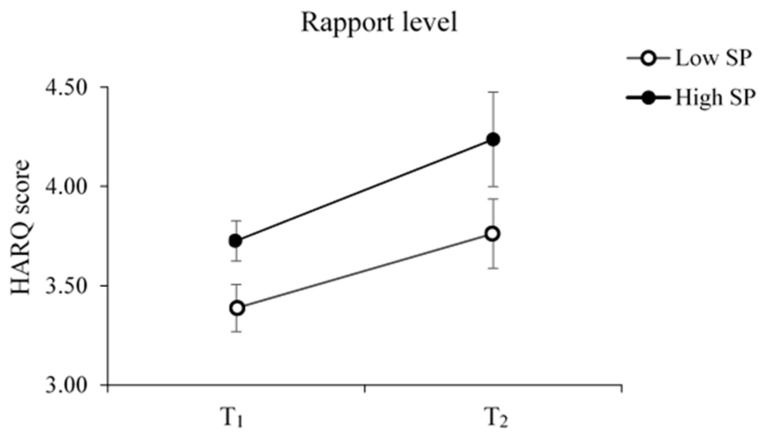
Effect of social presence and repeated CRIs on children–robot rapport levels. The CRIs positively fostered the children’s rapport with RoBoHon. Additionally, the High SP group reported marginally higher rapport than the Low SP group throughout the repeated CRI sessions. Error bars represent one standard error from the mean.

**Figure 4 sensors-23-04231-f004:**
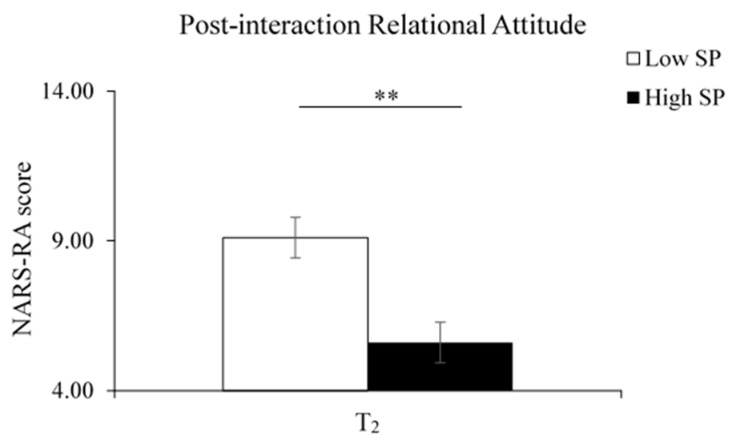
Social presence on post-interaction RA (T_2_) after controlling for pre-interaction RA (T_0_). The High SP group showed significantly reduced relational concerns with robots. Error bars represent one standard error from the mean. ** *p* < 0.01.

## Data Availability

The data and materials for all experiments can be found at https://osf.io/gj2yc/ (accessed on 1 March 2023).

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
