# Peer review of "The Effects of Social Presence and Familiarity on Children–Robot Interactions"

_sensors, 2023, doi:10.3390/s23094231_

Round 1

Reviewer 1 Report

This research topic is not novel, and it is not known what industrial effects the research results obtained through the interaction between children and robots help the education of kindergartens.

I don't know if this manuscript fully reflects the existing related studies. The authors should also clearly present differentiation and connectivity from existing related studies.

In addition, this manuscript is not considered to be related to scope of the journal. I think it is suitable for service robot journals or HRI journals.

Author Response

Dear Reviewer,

Thank you for taking the time to review our revised manuscript, “The effects of social presence and familiarity on children-robot interactions.” We appreciate the valuable feedback from the reviewers and are pleased to inform you that we have addressed all the concerns raised in the review.

Please see the attachment for the point-by-point response to each of the comments.

Reviewer 2 Report

 Thank you for the opportunity to review this well written and interesting manuscript.

Abstract

It could help the reader better understand the study from the abstract by adding some key details, like sample size, demographics (how old were these children), the context of the study (setting, situation).

Introduction

The introduction is well written, well structured and summarises relevant literature well while justifying the need for this study.

Methods

Methods are well described and replicable.

Results/Discussion

The results section and discussion are well written and provide useful and interesting contributions to the field.

Perhaps discuss implications of the recruited children already being in attendance at a robot workshop, perhaps suggesting prior interest in robots or greater acceptability. Can be discussed in the context of generalisability.

Summary

Originality – as the authors demonstrate, there is significant research available on human-robot interaction, including for children, however much of the available literature is limited by short interactions and novelty effects. The authors respond here by assessing social present, familiarity and rapport over a series of interactions spanning four days. The paper is well written and synthesises relevant prior work.

Author Response

(The authors gave the same response as above.)

Reviewer 3 Report

This article gave me more insight into how children who interacted with the small RoBoHon humanoid robot several times during a robot camp developed 'rapport' with it. The decision to study childrens' interactions over days or weeks rather than in one session is significant (and overdue). In the future, this could be extended further to give attention to the uses of the robot in everyday life more than in the clinical setting. 

I'm interested in the validity of the rapport metaphor, as 'familiarity' may be a better term as it could apply equally to a social and a technical object. It seems that the anthropomorphic frame could be unnecessary or presumptuous. One of the other tests – 'social presence' – did not change over time during the study, suggesting that rapport may not be a social variable after all. 

Yet, this is an interesting and worthy study. I appreciate some of the caveats about the need to study robots in different social contexts because, as a media studies scholar, I take issue with some of the epistemological assumptions and methodological approaches in HRI more broadly (this is not particular to this paper, which sits within this tradition well). Having said that, I particularly appreciate some of the observations about the significance of technical features such facial recognition to identify people and address them individually, as to me this suggests that there are some features that begin reaching a threshold point of mediated social presence. 

I like how the paper manages the distinction between utility and relationship, instrumentalism and companionship, and the significance of users' expectations. To me, there is a literacy, cultural capital or digital capital involved in establishing expectations about social robots. As robots are consumer electronics as much as social agents, I believe that people (including children) understand them differently depending on their technical aptitude or knowledge as much as their openness to interaction with robots or positive or negative expectations. These are my intuitive judgements that would be interesting to research. 

I found some typos worth fixing: 

Page 3 line 102 'stopped using the robot soon as they learned' should be 'stopped using the robot as soon as they learned…'

Page 9 line 366 'with the robot soon as they found' should be 'with the robot as soon as they found'

Page 9 line 384 'analogous to the beginning of a friendship without many benefits considerations' should be something like 'analogous to the beginning of a friendship when friends are not conscious of the benefits of the relationship'. (Be careful here because 'friends with benefits' is an idiom that refers to friendships involving sex without commitment) 

Page 11 line 465: 'Our study has shown that such variations can create some children a head start in building rapport with a robot.' should be Our study has shown that such variations can give some children a head start in building rapport with a robot.

Page 12 line 498: 'Rapporte' should be 'Rapport'

Author Response

(The authors gave the same response as above.)

Round 2

Reviewer 1 Report

There are many papers on the role and relationship of robots in child-robot interactions. 

These papers are rarely cited properly. There are only papers on English assistive robots such as Kanda et al. (2004), 

but lacks reviews of appropriate similar topics.

In this paper, what activities did the children do for 5 days with robots? 

What activities did the children do specifically and how did it affect their social relationships? 

Did the children learn robot programming? Did the kids joke and chat through the robot's NLP module? 

What activities of children and robots can be called social? 

How can you measure social relationships in approximately 15 minutes of cumulative time?

 How does knowing about negative attitudes toward robots contribute without an accurate description of the activity? 

And is it a generalized research result according to the type of robot? 

I can't agree if the topic of this paper is novel, if there isn't enough research, or if I don't agree with the purpose of the study.

------some related references-------------------------------------

Kindergarten Social Assistance Robot (KindSAR) for Geometric Thinking and Metacognitive Development in Kindergarten Education: A Pilot Study, Guy Keren, Marina Fridin

Understanding the Behavior and Role of Social and Adaptive Robots in Education: A Teacher's Perspective

MI Ahmad, O Mubin, J Orlando - ... of the Fourth International Conference ..., 2016 - dl.acm.org

Embrace social-assisted humanoid robots by early childhood and elementary school teachers

M Fridin, M Belokopytov - the computer of human behavior, 2014 - Elsevier

A Study on Children's Perception of Play with Augmented Reality

J Han, M Jo, E Hyun, HJ So HJ - Educational Technology Research and …, , 2015 - Springer

Relationship between user experience and children's perception of educational robots

Hyun Hyo Yoon-son - 2010 5th ACM/IEEE International …, , 2010 - ieeexplore.ieee.org

Characteristics of Children's Playtime Robot Utilization: A Case Study

 - RO-MAN 2009-The 18th IEEE International …, , 2009 - ieeexplore.ieee.org

Intelligent Robot Content for Children with Language Disabilities

H Lee Yi - hyeon - educational technology and social studies, 2015 - JSTOR

Author Response

We appreciate you and the reviewers for your precious time reviewing our paper and providing valuable comments. The valuable and insightful comments led to possible improvements in the current version.  

Please see the attachment for point-by-point responses.
Thank you.
